# Effectiveness of Inactivated Vaccine against SARS-CoV-2 Delta Variant Infection in Xiamen, China—A Test-Negative Case-Control Study

**DOI:** 10.3390/vaccines11030532

**Published:** 2023-02-23

**Authors:** Tingjuan He, Meixia Wang, Hongfei Mi, Liansheng Xu, Wenkui Lu, Xue Ouyang, Zhinan Guo, Chenghao Su

**Affiliations:** 1School of Public Health, Xiamen University, Xiamen 361102, China; 2Department of Infection Control, Xiamen Clinical Research Center for Cancer Therapy, Xiamen 361015, China; 3Xiamen Center for Disease Control and Prevention, Xiamen 361021, China; 4Tong’an District Center for Disease Control and Prevention, Xiamen 361021, China

**Keywords:** COVID-19, Delta variant, vaccine effectiveness, test-negative case–control design

## Abstract

Objective: Vaccine effectiveness can measure herd immunity, but the effectiveness of inactivated vaccines in Xiamen remains unclear. Our study was designed to understand the herd immunity of the COVID-19 inactivated vaccine against the SARA-CoV-2 Delta variant in the real world of Xiamen. Methods: We carried out a test-negative case-control study to explore the vaccine’s effectiveness. Participants aged over 12 years were recruited. A logistic regression was used to estimate the odds ratio (OR) of the vaccine among cases and controls. Results: This outbreak began with factory transmission clusters, and spread to families and communities during the incubation period. Sixty percent of cases were confirmed in a quarantine site. A huge mass of confirmed cases (94.49%) was identified within three days, and nearly half of them had a low Ct value. Following an adjustment for age and sex, a single dose of inactivated SARS-CoV-2 vaccine yielded the vaccine effectiveness (VE) of the overall case, of 57.01% (95% CI: −91.44~86.39%), the fully VE was 65.72% (95% CI: −48.69~88.63%) against COVID-19, 59.45% against moderate COVID-19 and 38.48% against severe COVID-19, respectively. The VE of fully vaccinated individuals was significantly higher in females than in males (73.99% vs. 46.26%). The VE among participants aged 19~40 and 41~61 years was 78.75% and 66.33%, respectively, which exceeds the WHO’s minimal threshold. Nevertheless, the VE in people under 18 and over 60 years was not observed because of the small sample size. Conclusions: The single-dose vaccine had limited effectiveness in preventing infection of the Delta variant. The two doses of inactivated vaccine could effectively prevent infection, and clinical mild, moderate, and severe illness caused by the SARS-CoV-2 Delta variant in people aged 18–60 years in the real world.

## 1. Introduction

The coronavirus disease 2019 (COVID-19), caused by the severe acute respiratory syndrome coronavirus 2 (SARS-CoV-2) spread around the world three years ago. Several variants with different characteristics emerged, with higher transmissibility, greater virulence, and a gradually reduced pathogenicity compared with the original variants [1]. With diverse transmission capacities of different variants, the cumulative number of infections rose dramatically. As of 20 November 2022, more than 633 million infections and 6.5 million deaths were reported globally [2]. COVID-19 data from the World Health Organization (WHO) showed that for each new variant, there was a spike in the death rate caused by COVID-19. The COVID-19 epidemic has brought great challenges to public health security, social stability, and economic development. The Delta variant was first identified in India in October 2020 [3]. It was significantly more virulent than the original strain and other variants, and it became the mainstream variant that caused the peak of the new wave of outbreaks in countries around the world in 2021 [4,5]. Non-drug interventions are the most effective epidemic prevention and control measures when specific therapeutic drugs are not widely available from previous pandemics [6,7]. Vaccination is the most economical and effective public health intervention to prevent and control infectious diseases. 

According to the latest data, 12.943 billion doses of vaccines have been administered globally, and 5.444 billion people have received at least a single-dose vaccine [2]. With the exception of some countries in Africa, global vaccination rates have exceeded 100% (total administered doses per 100 people) [2]. Vaccine effectiveness (VE) is a measure of how well vaccines work in the real world. Effectiveness is measured by observing how well the vaccines work to protect communities as a whole. Scientists have assessed the vaccine’s effectiveness in the real world after the vaccine was marketed. Since the Delta variant appeared, some studies have shown that the effectiveness of various vaccines in preventing Delta variant infection has been reduced to varying degrees [8,9,10]. However, studies in Europe and the United States indicated that two doses of the vaccine are still effective in preventing hospitalization [11,12,13,14]. We experienced multiple invasions of the Delta variant in 2021, and the vaccine effectiveness of different doses of the inactivated vaccine in different age or gender groups has seldom been declared from some regions. Previous studies suggested that inactivated vaccine had a good protective effect on clinical severe diseases, but the herd immunity in Xiamen is still unclear.

The epidemic in Xiamen was caused by an imported confirmed case from Putian, Fujian Province. All close contacts were immediately quarantined for medical observation, and monitoring was continued for 14–21 days. By September 2021, Xiamen had administered over 8,135,600 doses, and the vaccination rate of the first and second doses reached 85.24% and 80.58%, respectively [15]. As such, these conditions have provided a rare opportunity to conduct a real world evaluation of the effectiveness of the SARS-CoV-2 inactivated vaccine against Delta variant infection in Xiamen.

## 2. Methods

### 2.1. Study Design 

In this study, a test-negative case-control design (TND) approach was used to evaluate the effectiveness of a COVID-19 inactivated vaccine. This trial design evolved from the test-negative design method (test-negative design), which was originally used to study the effectiveness of pneumococcal vaccines [16]. It has been applied to estimate the effectiveness of influenza virus vaccines since 2005 [17] And has since been widely used worldwide for vaccine effectiveness assessments. This approach can be utilized to explore the vaccine effectiveness against COVID-19 variants or to monitor changes in the vaccine effectiveness over time when large randomized control trials are not feasible. It can minimize the deviation of the case–control design.

### 2.2. Definition of Study Participants

All participants resided in Xiamen, a city with a population of nearly 5,000,000. We recruited participants aged 12 and older, given their priority to be vaccinated according to national policy as of August 2021, which was based on the advice from national and provincial COVID-19 vaccination work deployment guidance groups.

#### 2.2.1. Case and Control Definition

Participants were classed into two groups: SARS-CoV-2 test-positive cases (cases) and test-negative cases (controls). In the statement published by the WHO on the confirmation of COVID-19 cases, it is clear that the explicit sequence of the COVID-19 nucleic acid is mainly detected by a reverse transcriptase polymerase chain reaction (RT−PCR) [18]. In our study, the SARS-CoV-2 cases were tested using a RT-PCR, and the specimen was identified as a confirmed case of COVID-19 if the result was positive. COVID-19 cases were classified into mild, moderate, severe, and critical illnesses according to the latest edition (trial version 8) of the national diagnosis and treatment protocol for COVID-19 in China [19]. (1) Mild COVID-19 denoted cases without signs of pneumonia upon chest imaging; (2) moderate COVID-19 cases were classified with fever, respiratory symptoms, and imaging characteristics of pneumonia; (3) severe COVID-19 met any of the following criteria: shortness of breath, respiratory rate >30/min, resting oxygen saturation <93%, oxygenation index < 300 mmHg, clinical symptoms worsening progressively and lung imaging showing the obvious progression of lesions >50% within 24~48 h; (4) critical COVID-19 cases were those with respiratory failure requiring mechanical ventilation, shock or organ failure requiring admission to the ICU.

Test-negative controls were the close contacts during the same period with a negative nucleic acid test (the Ct value was over 40 by RT-PCR). In China’s COVID-19 prevention and control measures (eighth edition) [20], the determination time of close contacts was advanced to 4 days before the onset of symptoms of confirmed patients, or from 4 days before sampling of asymptomatic patients in our study, according to the characteristics of the Delta variant. According to the prevention and control plan and the actual situation, people who have the following conditions with confirmed patients are judged as close contacts: (1) living in the same room; (2) close contact in the same environment; (3) sharing meals, entertainment and providing catering and entertainment services in a confined environment; (4) using the same vehicle and having close contact with people (within 1 meter); (5) exposure to the environment and objects contaminated by a case or an asymptomatic infected person.

#### 2.2.2. Definition of Immune Status

According to the results of phase 3 clinical trials of an inactivated COVID-19 vaccine, at least 14 days were required to form the protective effects against SARS-CoV-2 infections [21,22]. To determine the immune status of the population, we collected the precise time of each vaccine dose for the subjects and divided the participants into three types according to the 14-day requirement after vaccination: unvaccinated, partially vaccinated, and fully vaccinated. We defined the first-dose vaccination (partially vaccinated) and second-dose vaccination (fully vaccinated), 14 days having elapsed following the first dose or second dose, upon a clinical diagnosis (for test−positive cases) or when the first case was imported into Xiamen (for contacts). Otherwise, subjects would be deemed unvaccinated, despite the fact that they had received the first dose of vaccination.

### 2.3. Information Collection

The epidemiological investigation results, clinical classification, activity trajectory, medical visit records, close contact tracing, and other information on confirmed cases were all obtained from the confirmed cases migrant report completed by the migrant transfer staff of the Xiamen Center for Disease Control and Prevention during the pandemic period. Some of the data were further verified and supplemented by the Xiamen COVID-19 surveillance and traceability system integrated platform. Vaccination information of confirmed cases and close contacts was collected from the Xiamen Immunization Planning and Management cloud platform, which included the information of all people who had received COVID-19 vaccines in Xiamen, along with the date of administration and vaccine brand.

### 2.4. Statistical Analysis

Categorical and continuous variables were described by the frequency (%), and means (±standard deviation, SD), respectively. The categorical and continuous variables were compared between groups by using the Mann–Whitney U test, chi−squared test, or the Fisher exact test, as appropriate. A logistics regression model with infection status as dependent variables was used to evaluate the odds ratio (OR) of different vaccination statuses. The vaccine effectiveness was estimated by *VE* = (1 − *OR*) × 100%, which had been adjusted for age and gender. Subgroup analyses by stratification according to age, sex, and clinical severity were performed. The level of statistical significance was defined as *p* < 0.05 with a two-sided Anova. All data were analyzed using R Statistical software 4.1.3.

## 3. Results

### 3.1. Study Participant Recruitment

A total of 236 test−positive cases and 5092 close contacts with the negative nucleic acid test were confirmed in Xiamen and 41 confirmed cases and 1212 close contacts were excluded. The exclusion criteria are shown in Figure 1. In the end, we included 195 test−positive cases and 3840 test−negative controls in the final analysis of the vaccine effectiveness. The trial profile is shown in Figure 1.

### 3.2. Characteristics of Cases

#### Description of Epidemiological and Clinical Characteristics

This clustered outbreak is linked to an imported case from another city, lasting for about 20 days from September 12 to October 2, 2021. A total of 236 cases were reported, and most confirmed SARS-CoV-2 cases (81.78%) were aged 18~60 years. It showed that in 69.92% of cases, those infected had been fully vaccinated with two doses of vaccines. Among these, in 82.2% of cases, the individuals had received inactivated COVID-19 vaccines (Table 1). More than half of the cases (58.47%) were contracted from close contact in a quarantine site; 24.58% of cases were detected by nucleic acid screening in Tongan District (high-risk regions); 43.22% of cases were contracted in a factory setting, which was the main transmission method in this cluster. Their after-work activities resulted in family and social transmissions. In approximately half of the cases (46.61%), symptoms were reported within three days after the first viral RNA detection, and 47.46% of confirmed patients had a low Ct value at the first nucleic acid testing, indicating a high viral load. There were statistically significant differences between the vaccination groups with different clinical severities among the 236 confirmed cases. Vaccinated patients had a lower frequency of clinical severity than unvaccinated patients, especially in terms of critical illnesses (0.00% vs. 3.03%, *p* = 0.012). The differences in the Ct value, time from the onset of symptoms to nucleic acid testing, transfer times, and duration of hospital stays between different vaccination statuses are presented in Figure 2.

### 3.3. SARS-CoV-2 Vaccine Effectiveness

#### 3.3.1. Characteristics of Cases and Controls

According to the principles of the test-negative case-control design, the study included 195 cases from the positive test group and 3840 cases from the negative control group. Demographic characteristics and vaccination status are provided in Table 2. The age of the case group was significantly older than the control group (median: 40 vs. 36 years, *p* < 0.01). Participants between 40~60 years of age made up 49.23% of cases in the case group, and 50.86% of individuals aged 18~40 years were in the control group. The proportion of different genders reached a balance, which meant that there was no significant difference between the two groups. The clinical severity had the highest proportion of moderate illnesses (76.4%), and the only critical case in this outbreak was excluded from the analysis of the vaccine effectiveness due to a lack of vaccination information.

According to the previous clinical trials of the inactivated COVID-19 vaccine [23], protection against a SARS-CoV-2 infection was established 14 days after vaccination with the inactivated vaccine. There were 35 cases (17.95%) and 645 controls (16.80%) who had received one dose of the inactivated COVID-19 vaccine more than 14 days prior. Nearly 82.55% of the subjects had completed two doses of the inactivated COVID-19 vaccine, including 157 cases (80.51%) and 3174 controls (82.66%), for which there were no significant differences between the two groups (*p* = 0.190). The vaccine brands included BBIBP-CorV (Beijing Bio-chemicals, Wuhan Bio-chemicals, et al.) and CoronaVac vaccines, and 56.80% of subjects received the same brand of vaccine (BBIBP-CorV 17.72% and CoronaVac 39.08%) for both doses. Others (43.17%) received a combination of the two vaccines.

#### 3.3.2. SARS-CoV-2 Vaccine Effectiveness

A total of 4035 subjects were included in the study for the vaccine effectiveness analysis, and more than 90% of cases were of mild and moderate illnesses. For the whole population, the vaccine effectiveness of the partially vaccinated was 57.01% (95% CI: −91.44% to 86.39%), adjusted for age and sex, and the VE of the fully vaccinated was 65.72% (95% CI: −48.69% to 88.63%). The VE of fully vaccinated against mild and moderate COVID-19 cases exceeded 60%. However, the VE against severe cases (partially vaccinated as reference) was only 38.48% (95% CI: −314.82 to 85.24), which may be underestimated because of the small sample of the severe cases group (Table 3). The VE of different vaccinated statuses is presented in Figure 3.

We further analyzed the VE according to age group, gender, and clinical severity. It was found that the VE of the fully vaccinated was higher among females than that in males (73.99% vs. 46.26%). Noticeably, the VE against severe illnesses among females was more than twice that of males. In the age group, the effectiveness of the one-dose against the Delta variants reached 78.36% (95% CI: −2.03% to 93.94%) among subjects aged 19~40 years, and 78.75% (95% CI: 6.79% to 93.17%) of the fully vaccinated. In brief, the VE of the partially vaccinated and fully vaccinated in preventing non-severe COVID-19 cases approached 80%. No meaningful result was observed in patients under the age of 18 and over 60 because of the small samples.

## 4. Discussion

Vaccine effectiveness is an indication/indicator of herd immunity in the real world. People must take all of the required doses of a vaccine whose protection takes time to build. We developed and invested in a COVID-19 vaccine as quickly as possible. However, a highly infectious SARS-CoV-2 variant, B.1.617.2 (Delta), was identified and traced in Guangdong Province in late May 2021 [24,25]. Due to its stronger transmission and higher virulence, this variant caused a series of local outbreaks in China. These outbreaks provide the only opportunity for indigenous research on the vaccine effectiveness from the real world.

Our evaluation of the effectiveness of inactivated vaccines against the Delta variant demonstrated that the effectiveness against infection for the fully vaccinated was 65.72%, 75.97% against a mild illness, and 59.45% against a moderate illness. Our findings confirm the VE of inactivated vaccines against the Delta variant that have been reported by real world studies [13,26,27,28]. For instance, the effectiveness of the fully vaccinated against a SARS-CoV-2 infection was 51.8%, the effectiveness against a symptomatic illness was 60.4%, and the effectiveness for pneumonia was 78.4% in Guangdong Province in May-June 2021 [26]. Subsequent sporadic cases across the country supported studies of the vaccine effectiveness in different regions. Studies [13,27,28] have shown that after completing the basic immunization program with two doses, the VE for preventing symptomatic COVID-19 was over 50.00%, preventing pneumonia was above 60.00%, and against a severe clinical outcome, exceeded 80.00%, even up to 100%.

However, inactivated vaccines may not be equally effective against the B.1.617.2 variant, compared to the original variants. Of note, the VE was only 38.48% against severe illnesses in our study. We noticed that in nine severe cases, the individuals had finished their vaccination program more than two months earlier (mean time: 83 days). A study [29] indicated that protection remained high and stable during the first three months following the second dose but decreased slightly after the fourth month, especially in those over 60 days. Another study [30] found that the vaccine effectiveness of CoronaVac against a COVID-19 infection declined from 74.5% in the first two months to 30.4% after more than three months. Therefore, inactivated vaccines should be considered an option for immunity reinforcement programs upon completion of the population-level, two-dose vaccination program. Research in Chile has revealed that a booster vaccine obviously improves protection, the adjusted VE of three doses of CoronaVac against symptomatic COVID-19 was 78.8%, and against hospitalization and ICU admissions, it was 86.3% and 92.2% during the Delta outbreak [31].

At present, a number of studies on the effectiveness of COVID-19 vaccines based on the real world have been conducted in China. Our work contains the following strengths: first, the test-negative case–control design could reduce bias due to the differences in health-seeking behaviors and vaccination. It is currently the most effective and unbiased study design in the real world. Second, all of the cases detected in this outbreak were caused by the first case. It is a single source of infection and a clear transmission chain, which created a clean epidemiological study field. Thus, it can remove more confounding factors and decrease the admission rate bias (Berkson bias). 

However, there are still some limitations in this study. Although we controlled for the known covariate, residual unmeasured confounders may have affected the vaccine effectiveness in the analysis, as with all observational studies. Secondly, because of the lack of vaccination information on several subjects, we decided to exclude those people when evaluating the vaccine’s effectiveness. Another limitation is the small number of variables that undermined the possibility of a subgroup analysis. The second dose vaccination rates for people aged 12~18 and over 60 years were only 5.03% and 6.44%, respectively, until 6 September 2021.

In a word, our research showed that the vaccine effectiveness of a single dose of an inactivated vaccine is restricted. Two doses of the inactivated vaccine could effectively prevent a Delta variant infection, clinical mild, moderate, and severe illnesses in those aged 18~60 years, which reached the threshold value set by the WHO for the COVID-19 vaccine [32]. We believe that our study provided useful insights into the effectiveness of inactivated vaccines and suggested that being fully vaccinated is effective against the B.1.6.17.2 variant of SARA-CoV-2. We should continue to undertake large-scale booster vaccination programs for children and the elderly, to promote their immunity barrier. With booster and duration, vaccine effectiveness will be a key question for future investigation. Our next challenge is to evaluate the herd immunity and the effective duration in Xiamen.

## Figures and Tables

**Figure 1 vaccines-11-00532-f001:**
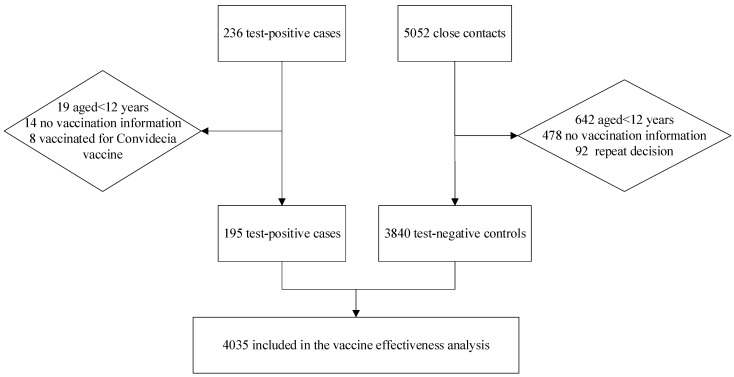
Flowchart of selection of study participants.

**Figure 2 vaccines-11-00532-f002:**
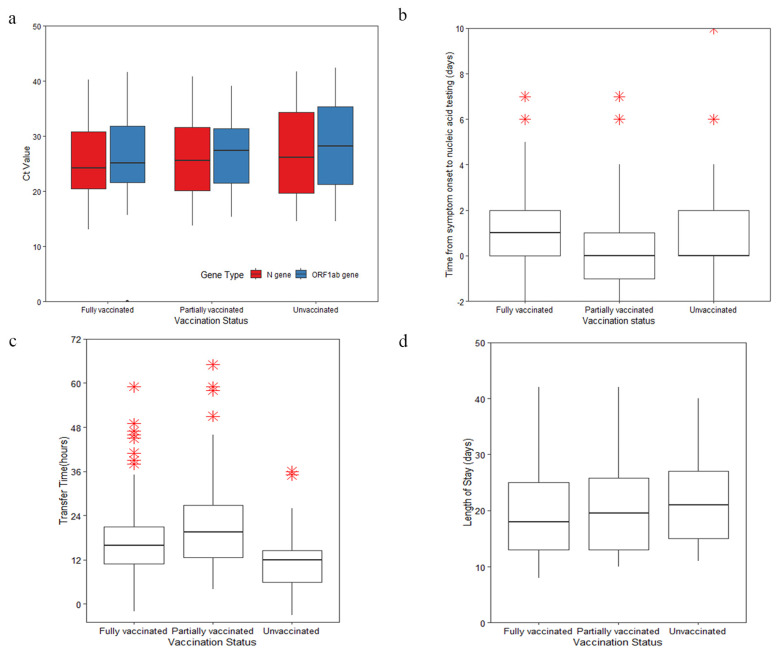
Epidemiological Characteristics of infected individuals. (**a**) Distribution of Ct values of positive nucleic acid test for ORF1ab and N gene in infected individuals by vaccination status; (**b**) Duration of time from symptom onset to nucleic acid testing in infected persons by vaccination status; (**c**) Duration of transfer time in infected persons by vaccination status; (**d**) Length of stay of infected individuals by vaccination status. Note: * means abnormal value.

**Figure 3 vaccines-11-00532-f003:**
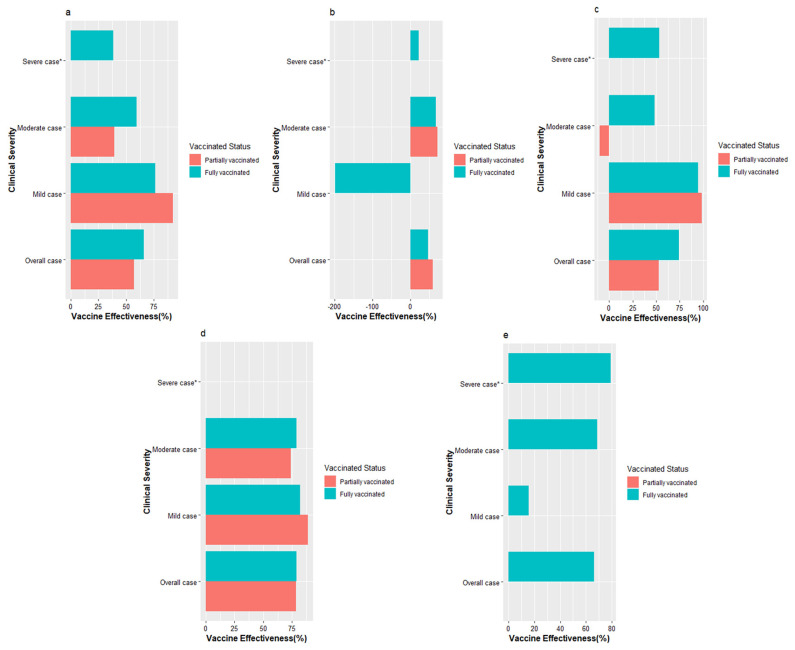
Effectiveness of the inactivated vaccines against different severity of COVID−19 associated with the Delta variant strain. (**a**) Effectiveness in the whole population. The abscissa is vaccine effectiveness (%) and the ordinate are different clinical severity. The bars represent the estimated value of VE. Different colors indicate different vaccination statuses; (**b**) Effectiveness in males; (**c**) Effectiveness in females; (**d**) Effectiveness in the study participants aged 19–40 years; (**e**) Effectiveness in study participants aged 41–60 years. Note: * partially vaccinated was used as reference because of lacking data on unvaccinated group.

**Table 1 vaccines-11-00532-t001:** Epidemiological Characteristics of all the cases identified in the outbreak in Xiamen.

	Overall [N (%)]	Unvaccinated [N (%)]	Vaccinated [N (%)]	*p* Value
Age (year)				
<=18	30 (12.71)	20 (60.61)	10 (4.93)	<0.001
18~60	193 (81.78)	8 (24.24)	185 (91.13)	
>60	13 (5.51)	5 (15.15)	8 (3.94)	
Gender				
male	116 (49.15)	21 (63.64)	95 (46.80)	0.108
female	120 (50.85)	12 (36.36)	108 (53.20)	
Vaccination status				
Unvaccinated	35 (14.83)	33 (100.00)	2 (0.99)	<0.001
Partially vaccinated	36 (15.25)	0 (0.00)	36 (17.73)	
Fully vaccinated	165 (69.92)	0 (0.00)	165 (81.28)	
Vaccine type				
Inactivated vaccine	194 (82.20)	0 (0.00)	194 (95.57)	<0.001
Ad5-nCoV vaccine	8 (3.39)	0 (0.00)	8 (3.94)	
Unvaccinated/Unknown	34 (14.41)	33 (100.00)	1 (0.49)	
Source				
Quarantine site test	138 (58.47)	22 (66.67)	116 (57.14)	0.236
High-risk regions screening	58 (24.58)	3 (9.09)	55 (27.09)	
Full screening	18 (7.63)	4 (12.12)	14 (6.90)	
Key population screening	6 (2.54)	1 (3.03)	5 (2.46)	
Initiative to see a doctor	16 (6.78)	3 (9.09)	13 (6.40)	
Type of epidemic transmission				
Factory transmission	102 (43.22)	5 (15.15)	97 (47.78)	<0.001
Family transmission	65 (27.54)	17 (51.52)	48 (23.65)	
Community transmission	69 (29.24)	11 (33.33)	58 (28.57)	
Time from symptom onset to nucleic acid testing (day)				
<=0	113 (47.88)	17 (51.52)	96 (47.29)	0.490
1~3	110 (46.61)	13 (39.39)	97 (47.78)	
>=4	13 (5.51)	3 (9.09)	10 (4.93)	
PCR cycle threshold (Ct value, copies/mL)				
<=24	112 (47.46)	14 (42.42)	98 (48.28)	0.371
24~35	88 (37.29)	11 (33.33)	77 (37.93)	
35~40	30 (12.71)	6 (18.18)	24 (11.82)	
negative	6 (2.54)	2 (6.06)	4 (1.97)	
ORF1ab gene (open reading frame 1ab)	26.78 (6.94)	28.45 (7.79)	26.51 (6.77)	0.137
N gene (nucleocapsid protein)	25.89 (7.18)	27.67 (8.32)	25.60 (6.96)	0.126
Transfer time(hours)	17.73 (11.97)	12.58 (9.53)	18.57 (12.14)	0.007
Clinical symptoms				
No	44 (18.64)	8 (24.24)	36 (17.73)	0.516
Yes	192 (81.36)	25 (75.76)	167 (82.27)	
Clinical severity				
Mild	50 (21.19)	11 (33.33)	39 (19.21)	0.012
Moderate	176 (74.58)	21 (63.64)	155 (76.35)	
Severe	9 (3.81)	0 (0.00)	9 (4.43)	
Critical	1 (0.42)	1 (3.03)	0 (0.00)	
Length of stay (day)	19.99 (8.07)	21.45 (7.36)	19.75 (8.17)	0.261

**Table 2 vaccines-11-00532-t002:** Characteristics of SARS-CoV-2 cases and controls for vaccine effectiveness estimation.

	Overall [N (%)]	Positive Cases [N (%)]	Negative Controls [N (%)]	*p* Value
Age group (years)				
<18	444 (11.00)	10 (5.13)	434 (11.30)	<0.001
18–40	2033 (50.38)	80 (41.03)	1953 (50.86)	
40–60	1427 (35.37)	96 (49.23)	1331 (34.66)	
>=60	131 (3.25)	9 (4.62)	122 (3.18)	
Gender				
Male	2074 (51.40)	90 (46.15)	1984 (51.67)	0.153
Female	1961 (48.60)	105 (53.85)	1856 (48.33)	
Clinical severity				
Mild	149 (3.69)	37 (18.97)	-	
Moderate	37 (0.92)	149 (76.41)	-	
Severe	9 (0.22)	9 (4.62)	-	
Vaccinated status				
Unvaccinated	24 (0.59)	3 (1.54)	21 (0.55)	0.190
Partially vaccinated	680 (16.85)	35 (17.95)	645 (16.80)	
Fully vaccinated	3331 (82.55)	157 (80.51)	3174 (82.66)	
Vaccine brand				
Unvaccinated	1 (0.02)	1 (0.51)	0 (0.00)	<0.001
BBIBP-CorV	715 (17.72)	36 (18.46)	679 (17.68)	
CoronaVac	1577 (39.08)	78 (40.00)	1499 (39.04)	
Mixed vaccine #	1742 (43.17)	80 (41.03)	1662 (43.28)	
Duration of vaccination until the outbreak * (days)				
	42.63 (45.06)	44.73 (44.42)	42.52 (45.10)	0.503
Total	4035	195	3840	

Note: # Two doses of BBIBP-CorV vaccine and CoronaVac vaccine were available; * The interval between 14 days after vaccination and the time when the first case of this outbreak was reported.

**Table 3 vaccines-11-00532-t003:** Estimates of the association between vaccination and SARS-CoV-2 infection and effectiveness of inactivated SARS-CoV-2 vaccines.

		Unvaccinated	Partially Vaccinated	OR_adj_ (95% CI)	VE_adj_ (%, 95% CI)	Fully Vaccinated	OR_adj_ (95% CI)	VE_adj_ (%, 95% CI)
The whole population	Control	21	645	Reference		3175	Reference	
	Overall case	3	35	0.43 (0.14, 1.91)	57.01 (−91.44, 86.39)	157	0.34 (0.11, 1.49)	65.72 (−48.69, 88.63)
	Mild case	1	3	0.08 (0.01, 1.68)	91.95 (−67.87, 99.04)	33	0.24 (0.05, 4.4)	75.97 (−339.82, 95.29)
	Moderate case	2	30	0.61 (0.16, 4.08)	39.09 (−308.39, 84.3)	117	0.41 (0.11, 2.66)	59.45 (−165.87, 89.06)
	Severe case *	0	2	-	-	7	0.62 (0.15, 4.15)	38.48 (−314.82, 85.24)
Male	Control	13	369	Reference		1602	Reference	
	Overall case	1	12	0.41 (0.07, 7.74)	59.11 (−674.18, 92.86)	77	0.54 (0.1, 9.86)	46.26 (−885.99, 89.63)
	Mild case *	0	2	-	-	20	2.99 (0.83, 19.18)	−199.25 (−1817.92, 16.54)
	Moderate case	1	9	0.29 (0.05, 5.53)	71.36 (−452.7, 95.29)	53	0.32 (0.06, 6.01)	67.53 (−501.04, 93.9)
	Severe case *	0	1	-	-	4	0.77 (0.11, 15.21)	22.79 (−1420.85, 88.72)
Female	Control	8	276	Reference		1572	Reference	
	Overall case	2	23	0.48 (0.1, 3.42)	52.47 (−241.68, 89.52)	80	0.26 (0.06, 1.81)	73.99 (−81.47, 93.95)
	Mild case	1	1	0.02 (0, 0.47)	98.34 (52.94, 99.94)	13	0.05 (0.01, 1.01)	94.85 (−1.21, 99.21)
	Moderate case	1	21	1.1 (0.17, 22.1)	−10.18 (−2110.26, 83.11)	64	0.52 (0.08, 10.22)	48.13 (−921.83, 91.66)
	Severe case	0	1	-	-	3	0.47 (0.06, 9.49)	53.21 (−848.76, 94.05)
12~18 years	Control	0	209	Reference		155	Reference	
	Overall case *	0	1	-	-	9	7.31 (1.35, 135.52)	−631.16 (−13451.89, −35.42)
	Mild case	0	0	-	-	6	-	-
	Moderate case *	0	1	-	-	3	2.32 (0.29, 47.16)	−131.83 (−4615.9, 70.7)
	Severe case	0	0	-	-	0	-	-
19~40 years	Control	16	285	Reference		1726	Reference	
	Overall case	3	12	0.22 (0.06, 1.02)	78.36 (−2.03, 93.94)	75	0.21 (0.07, 0.93)	78.75 (6.79, 93.17)
	Mild case	1	2	0.11 (0.01, 2.51)	88.71 (−150.5, 98.97)	19	0.18 (0.03, 3.31)	82.15 (−231.27, 96.67)
	Moderate case	2	10	0.26 (0.06, 1.79)	74.00 (−79.23, 93.9)	53	0.21 (0.06, 1.37)	78.78 (−37.4, 94.27)
	Severe case	0	0	-	-	3	-	-
41~60 years	Control	3	117	Reference		1118	Reference	
	Overall case *	0	21	-	-	66	0.34 (0.2, 0.58)	66.33 (41.86, 79.8)
	Mild case *	0	1	-	-	8	0.84 (0.15, 15.71)	15.71 (−1471.3, 84.75)
	Moderate case *	0	18	-	-	54	0.31 (0.18, 0.57)	68.59 (43.31, 81.86)
	Severe case *	0	2	-	-	4	0.21 (0.04, 1.53)	79.02 (−52.65, 95.95)
more than 60 years	Control	2	34	Reference		75	Reference	
	Overall case	0	1	-	-	7	4.17 (0.71, 79.5)	−317.39 (−7850.35, 28.53)
	Mild case	0	0	-	-	0	-	-
	Moderate case	0	1	-	-	7	3.68 (0.6, 71.43)	-268.44 (−7043.2, 40.14)
	Severe case	0	0	-	-	0	-	-

Note: * partially vaccinated was used as reference because of lacking data on unvaccinated group; the “-” indicated no data.

## Data Availability

The data presented in this study are available on request from the corresponding author. The data are not publicly available due to privacy and ethical restrictions.

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
