# Peer review of "Effectiveness of Inactivated Vaccine against SARS-CoV-2 Delta Variant Infection in Xiamen, China—A Test-Negative Case-Control Study"

_vaccines, 2023, doi:10.3390/vaccines11030532_

Round 1

Reviewer 1 Report

The paper needs significant improvement before it can be considered for publication. 

Introduction: Is incomplete and lacks references. Specifically, 

Line 36 is transmission power is not a real term

Lines 44-48 - Must be referenced

Paragraph starting on lien 50 - Must define dVE and how it is calculated

Line 53 Must define the different vaccines and how they have performed differently. This is not complete

Line 66 - number makes no sense with . or , ?

Line 67 Reference 

Section 2.1

Lines 150 to 155 This does not make sense. It must be rewritten. 

Section 2.2 This must be completely described, what is the national policy and how was it implemented?

Section 2.2.1

The methods must be completely described. This is not reproducible. What are mild, moderate and severe criteria?

Section 2.2.2

The section needs references for the vaccines and a better explanation of which vaccine is which and what rates they were included in the study

Section 2.4

This needs to be fully explained including the models and how variables were excluded. 

Table 1: This table needs a proper explanation to describe each of the variables, what does vaccinated mean (1 dose, 2 dose???), What is compared with the reported p value? This table is very unclear. In the second half of the table (CT values) there needs to be a measure of dispersion for each mean. 

Figure 1

Not clear what the box that started with 236 is or means, figure needs a better explanation in the heading

Section 3: It is impossible to review or understand the results until the methods are clarified.  

Author Response

Dear Reviewer,

  We feel great thanks for your professional review work on our article. As you are concerned, there are several problems that need to be addressed. According to your nice suggestions, we have made extensive corrections to our previous draft, the detailed corrections are listed below. Our response is given in the blue bold text and changes/additions to the manuscript are given by underlining.

Reviewer 2 Report

The article is well written, methods are clearly described and the manuscript is well organized and clear in content and conclusions.  I recommend this manuscript for publication.

Author Response

Dear Reviewer,

      On behalf of all the contributing authors, I would like to express our sincere appreciations of your letter and reviewers’ constructive comments concerning our article entitled “Effectiveness of Inactivated Vaccine Against SARS-CoV-2 Delta Variant Infection in Xiamen, China-A Test-Negative Case-Control Study (Manuscript No. 2171639)". These comments are all valuable and helpful for improving our article.

Reviewer 3 Report

Effectiveness of Inactivated Vaccine Against SARS-CoV-2 Delta Variant Infection in Xiamen, China-A Test-Negative Case-Control Study

The study seems an excellent attempt by authors to evaluate the vaccine efficacy of the SARS-CoV-2 inactivated vaccine against SARS-CoV-2 delta variants in Xiamen, China. I found the study is well designed and executed meticulously and the results are clearly presented. However, this manuscript warrants revision and correction in English for better reading. Following are the specific comments to further strengthen the present manuscript,

1.     In the abstract abbreviation, VE appeared suddenly (line 20), is this vaccine effectiveness as mentioned in the introduction (line 53)? Whether vaccine efficacy or vaccine effectiveness, please consider it to mention the complete form with an abbreviation on the first appearance to prevent readers from assuming or searching.

2.     In the introduction (line 33), brought about by caused by, please try to rewrite the sentence.

3.     In the introduction (line 66), the vaccinated number is 8.135,600 or 8,135,600 doses.

4.     Please rewrite line 140 in the result for better understanding.

5.     Consider increasing the font sizes in figure 2. Also, these are a,b,c and d subfigures but in the legend appeared as A,b,c and d. (A is in the capital) try to maintain uniform representation.

6.     In result section 2. Characteristics of cases, there is subheading 2.1 but no 2.2 or is this necessary?

7.     Page 8, is this table 3? Or 2? I think this is table 2 and table 3 is on page 11, correct this.

8.     Consider increasing the font sizes in figure 3, as it is hard to read.

9.     What, if any data is available to compare between fully vaccinated and recently booster dose administered subjects for VE against the delta variant?

Author Response

(The authors gave the same response as above.)
